# ADVERSARIAL DROPOUT REGULARIZATION

Kuniaki Saito[1], Yoshitaka Ushiku[1], Tatsuya Harada[1,2], and Kate Saenko[3]

[1]The University of Tokyo, [2]RIKEN, [3]Boston University
{k-saito,ushiku,harada}@mi.t.u-tokyo.ac.jp, saenko@bu.edu

## ABSTRACT

We present a domain adaptation method for transferring neural representations from label-rich source domains to unlabeled target domains. Recent adversarial methods proposed for this task learn to align features across domains by "fooling" a special domain classifier network. However, a drawback of this approach is that the domain classifier simply labels the generated features as in-domain or not, without considering the boundaries between classes. This means that ambiguous target features can be generated near class boundaries, reducing target classification accuracy. We propose a novel approach, Adversarial Dropout Regularization (ADR), which encourages the generator to output more discriminative features for the target domain. Our key idea is to replace the traditional domain critic with a critic that detects non-discriminative features by using dropout on the classifier network. The generator then learns to avoid these areas of the feature space and thus creates better features. We apply our ADR approach to the problem of unsupervised domain adaptation for image classification and semantic segmentation tasks, and demonstrate significant improvements over the state of the art.

## 1 INTRODUCTION

Transferring knowledge learned by deep neural networks from label-rich domains to new target domains is a challenging problem, especially when the source and target input distributions have different characteristics. Such *domain shifts* occurs in many practical applications. For example, while simulated driving images rendered by games provide a rich source of labeled data for semantic segmentation Richter et al. (2016), deep models trained on such source data do not transfer well to real target domains (Fig. 1(a-d)). When target-domain labels are unavailable for fine-tuning, unsupervised domain adaptation must be applied to improve the source model.

Recent methods for unsupervised domain adaptation attempt to reduce the discrepancy between the source and target features via adversarial learning (Tzeng et al. (2014); Ganin & Lempitsky (2014)). They divide the base network into a feature encoder $G$ and classifier $C$, and add a separate domain classifier (critic) network $D$. The critic takes the features generated by $G$ and labels them as either source- or target-domain. The encoder $G$ is then trained with an additional adversarial loss that maximizes $D$'s mistakes and thus aligns features across domains.

However, a major drawback of this approach is that the critic simply predicts the domain label of the generated point and does not consider category information. Thus the generator may create features that look like they came from the right domain, but are not discriminative. In particular, it can generate points close to class boundaries, as shown in Fig. 1(e), which are likely to be misclassified by the source model. We argue that to achieve good performance on the target data, the adaptation model must take the decision boundaries between classes into account while aligning features across domains (Fig. 1(f)). Moreover, since our setting is unsupervised adaptation, this must be accomplished without labels on target data.

In this paper, we propose a novel adversarial alignment technique that overcomes the above limitation and preserves class boundaries. We make the following observation: *if the critic could detect points near the decision boundary, then the generator would have to avoid these areas of the feature space in order to fool the critic.* Thus the critic would force the generator to create more discriminative features. How can we obtain such a critic? If we alter the boundary of the classifier $C$ slightly and measure the change in the posterior class probability $p(y|x)$, where $y$ and $x$ denote class and

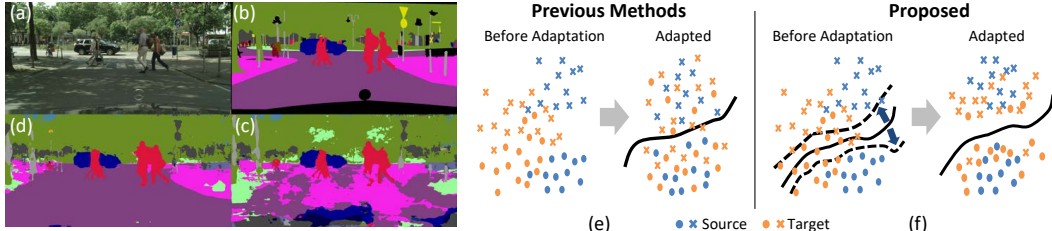

Figure 1: (a-d) An illustration of a deep model trained on simulated source training data failing to segment a real target domain image: (a) shows the target image, (b) is the ground truth segmentation into semantic categories (car, road, etc), (c) is the output of the unadapted source model, (d) is the improved segmentation obtained by our proposed ADR method. (e) Previous distribution matching methods do not consider the source decision boundary when aligning source and target feature points. (f) We propose to use the boundary information to achieve low-density separation of aligned points.

input respectively, then samples near the decision boundary are likely to have the largest change. In fact, this posterior discrepancy is inversely proportional to the distance from the class boundary. We thus propose to maximize this posterior discrepancy to turn $C$ into a critic sensitive to non-discriminative points. We call this technique *Adversarial Dropout Regularization*. Here, dropout is not used in the standard way, which is to regularize the main classifier and make it insensitive to noise. Instead, we use dropout in an adversarial way, to transform the classifier into a critic sensitive to noise. Compared to previous adversarial feature alignment methods, where the distributions $p(x)$ are aligned globally, our method aligns target features away from decision boundaries, as illustrated in Fig.1(f).

Our ADR approach has several benefits. First, we train the generator $G$ with feedback from the classifier $C$, in contrast to existing methods, which use an unrelated critic $D$. Second, our method is general and straightforward to apply to a variety of domain adaptation problems, such as classification and semantic segmentation. Finally, since ADR is trained to align distributions, it is also applicable to semi-supervised learning and training of generative models, such as Generative Adversarial Networks (GANs) (Goodfellow et al. (2014a)). Through extensive experiments, we demonstrate the benefit of ADR over existing domain adaptation approaches, achieving state-of-the-art results in difficult domain shifts. We also show an application to semi-supervised learning using GANs in appendix.

## 2 RELATED WORK

**Domain Adaptation.** Recent unsupervised domain adaptation (UDA) methods for visual data aim to align the feature distributions of the source and target domains (Sun et al. (2016); Sun & Saenko (2016); Tzeng et al. (2014); Ganin et al. (2016); Long et al. (2015b); Yan et al. (2017); Long et al. (2017)). Such methods are motivated by theoretical results stating that minimizing the divergence between domains will lower the upper bound of the error on target domain (Ben-David et al. (2010)). Many works in deep learning utilize the technique of distribution matching in hidden layers of a network such as a CNN (Tzeng et al. (2014); Ganin et al. (2016); Long et al. (2015b)). However, they measure the domain divergence based on the hidden features of the network without considering the relationship between its decision boundary and the target features, as we do in this paper.

**Low-density Separation.** Many semi-supervised learning (SSL) methods utilize the relationship between the decision boundary and unlabeled samples, a technique called low-density separation (Chapelle & Zien (2005); Joachims (1999)). By placing the boundary in the area where the unlabeled samples are sparse, these models aim to obtain discriminative representations. Our method aims to achieve low-density separation for deep domain adaptation and is related to entropy minimization for semi-supervised learning (Grandvalet & Bengio (2005)). (Long et al. (2016)) used entropy minimization in their approach to directly measure how far samples are from a decision boundary by calculating entropy of the classifier's output. On the other hand, our method tries to achieve low-density separation by slightly moving the boundary and detecting target samples sensitive to the movement. As long as target samples features are robust to the movement, they will be allowed to exist relatively nearby the boundary compared to source samples, as Fig. 1 shows.

In (Long et al. (2016)) entropy minimization is only a part of the overall approach. To compare our ADR approach to entropy minimization more directly, we use a new baseline method. To our knowledge, though this method has not been proposed by any previous work, it is easily achieved by modifying a method proposed by (Springenberg (2015)). For this baseline, we train a model that generates features to minimize the entropy of the output probability for target samples. The details of the baseline are provided in appendix. In short, the generator tries to minimize the entropy of the target samples, whereas the critic tries to maximize it. The entropy is directly measured by the output of the classifier. This baseline is similar to our approach in that the goal of the method is to achieve low-density separation.

**Dropout.** Dropout is a method that prevents deep networks from overfitting (Srivastava et al. (2014)) by randomly dropping units from the neural network during training. Effectively, dropout samples from an exponential number of different thinned networks at training time, which prevents units from co-adapting too much. At test time, predictions are obtained by using the outputs of all neurons. If the thinned networks are able to classify the samples accurately, the full network will as well. In other words, dropout encourages the network to be robust to noise. In our work, we use dropout to regularize the feature generation network $G$, but in an adversarial way. We train the critic $C$ to be sensitive to the noise caused by dropout and use $C$ to regularize $G$ so that it generates noise-robust features. To our knowledge, this use of dropout is completely different from existing methods.

## 3 METHOD

We assume that we have access to a labeled source image $\mathbf{x_s}$ and a corresponding label $y_s$ drawn from a set of labeled source images $\{X_s, Y_s\}$, as well as an unlabeled target image $\mathbf{x_t}$ drawn from unlabeled target images $X_t$. We train a feature generation network $G$, which takes inputs $\mathbf{x_s}$ or $\mathbf{x_t}$, and a network $C$ that acts as both the main classifier and the critic. When acting as the classifier, $C$ takes features from $G$ and classifies them into $K$ classes, predicting a $K$-dimensional vector of logits $\{l_1, l_2, l_3...l_K\}$. The logits are then converted to class probabilities by applying the softmax function. Namely, the probability that $\mathbf{x}$ is classified into class $j$ is denoted by $p(y = j|\mathbf{x}) = \frac{exp(l_j)}{\sum_{k=1}^{K} exp(l_k)}$. We use the notation $p(\mathbf{y}|\mathbf{x})$ to denote the $K$-dimensional probabilistic output for input $\mathbf{x}$.

When $C$ is acting as the critic, we want it to detect the feature encodings of target samples near the decision boundary. We propose to make $C$ sensitive to such samples by slightly perturbing its decision boundary and measuring the change in the posterior class probability $p(\mathbf{y}|\mathbf{x})$. This change is likely to be largest for samples near the decision boundary. The network $C$ is then trained to increase this change, while the feature generation network $G$ is trained to decrease it. Through this adversarial training, $G$ learns to 'fool' the critic and generate target features far away from the decision boundary, thus avoiding ambiguous features. The weights of $G$ can be initialized either by pre-training on some auxiliary dataset (e.g., ImageNet), or with random weights, while $C$ uses random initialization. In the next section, we show how we utilize dropout to perturb the boundary in the critic and measure sensitivity. We then show the training procedure of our method. Finally, we give some intuition behind adversarial dropout and improve our method based on this insight.

### 3.1 CLASSIFIER SELECTION VIA DROPOUT

Consider the standard training of a neural network using dropout. For every sample within a mini-batch, each node of the network is removed with some probability, effectively selecting a different classifier for every sample during training. We harness this idea in a very simple way.

We forward input features $G(\mathbf{x_t})$ to $C$ twice, dropping different nodes each time and obtaining two different output vectors denoted as $C_1(G(\mathbf{x_t}))$, $C_2(G(\mathbf{x_t}))$. In other words, we are selecting two different classifiers $C_1$ and $C_2$ from $C$ by dropout as in Fig. 2. In the figure, the corresponding posterior probabilities are indicated as $p_1(\mathbf{y}|\mathbf{x_t})$, $p_2(\mathbf{y}|\mathbf{x_t})$, abbreviated as $p_1$ and $p_2$ in the following discussion. In order to detect the change of predictions near the boundary, the critic tries to increase the difference between the predictions of $C_1$ and $C_2$. This difference corresponds to $C$'s sensitivity to the noise caused by dropout.

To measure the sensitivity $d(p_1, p_2)$ between the two obtained probabilistic outputs, we use the symmetric Kullback Leibler (KL) divergence. Formally, the divergence is calculated as

$$d(p_1, p_2) = \frac{1}{2}(D_{kl}(p_1|p_2) + D_{kl}(p_2|p_1)) \tag{1}$$

Figure 2: Overview of ADR. **Left**: We train $G$, $C$ with classification loss on source and sample a critic consisting of two classifiers using dropout. The critic's sensitivity is measured as the divergence between the class predictions of $C_1$ and $C_2$ on the same input. **Right**: Adversarial training iterates two steps: the critic tries to maximize the sensitivity while the generator tries to minimize it.

where KL divergence between $p$ and $q$ is denoted as $D_{kl}(p|q)$.

## 3.2 TRAINING PROCEDURE

In our approach, $C$ works as both critic and classifier. The following three requirements are imposed by our method: 1) $C$ and $G$ must classify source samples correctly to obtain discriminative features; 2) $C$ should maximize the sensitivity for target samples to detect the samples near the boundary; 3) $G$ should learn to minimize the sensitivity to move target samples away from the boundary.

The training within the same mini-batch consists of the following three steps.

**Step 1**, in this step, $C$ is trained as a classifier. $C$ and $G$ have to classify source samples correctly to obtain discriminative features. Thus, we update both networks' parameters based on the following standard classification loss. Given source labels $y_s$ and samples $\mathbf{x_s}$, the objective in this step is

$$\min_{G,C} L(X_s, Y_s) = -\mathbb{E}_{(\mathbf{x_s}, y_s) \sim (X_s, Y_s)} \sum_{k=1}^{K} \mathbb{1}_{[k=y_s]} \log C(G(\mathbf{x_s}))_k \qquad (2)$$

$C(G(\mathbf{x_s}))_k$ returns the probability that the sample $\mathbf{x_s}$ is assigned to class $k$.

**Step 2**, in this step, $C$ is trained as a critic to detect target samples near the boundary. Two classifiers are sampled from $C$ for each target sample using dropout twice to obtain $p_1$ and $p_2$. Then, $C$'s parameters are updated to maximize the sensitivity as measured by Eq. 1. Since $C$ should learn discriminative features for source samples, in addition to the sensitivity term, we add Eq. 2. We experimentally confirmed that this term is essential to obtain good performance.

$$\min_{C} L(X_s, Y_s) - L_{adv}(X_t) \qquad (3)$$

$$L_{adv}(X_t) = \mathbb{E}_{\mathbf{x_t} \sim X_t}[d(C_1(G(\mathbf{x_t})), C_2(G(\mathbf{x_t})))] \qquad (4)$$

$C_1$ and $C_2$ are sampled from $C$ randomly.

**Step 3**, in order to obtain representations where target samples are placed far from the decision boundary, $G$ is trained to minimize sensitivity. Here we do not add the categorical loss for source samples as in Step 2, as the generator is able to obtain discriminative features without it.

$$\min_{G} L_{adv}(X_t) \qquad (5)$$

We update the parameters of $C$ and $G$ in every step following the defined objectives. We experimentally found it beneficial to repeat Step 3 $n$ times for each mini-batch.

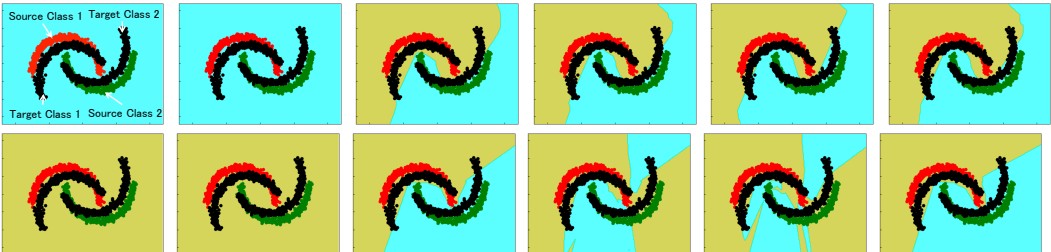

Figure 3: (Best viewed in color) Toy Experiment. **Top row**: Model trained without adaptation. Columns 1-5 show the decision boundary obtained by keeping one neuron in the last hidden layer and removing the rest. **Red points** are source samples of class one, **green points** are class two. **Black points** are target samples. **The yellow region** indicates where the samples are classified as class one, **cyan region** class two. We see that the neurons do not learn very diverse features. Column 6 shows the boundary obtained by keeping all 5 neurons. **Bottom row**: Boundaries learned by the model adapted by our adversarial dropout method. Unlike the top row, here neurons 3,4,5 learn diverse features which result in diverse boundaries.

### 3.3 INSIGHT AND IMPROVEMENT

Our ADR approach encourages different neurons of the classifier to learn different characteristics of the input (see Sec. 4.1.) The output is the combination of shared and unshared nodes, therefore, to maximize the sensitivity, the unshared nodes must learn different features of target samples. As learning proceeds, each neuron in $C$ will capture different characteristics. At the same time, to minimize the sensitivity, $G$ learns to extract pure categorical information. If $G$ outputs features which are not related to categorical information, such as texture, slight contrast or difference of color, $C$ will utilize them to maximize sensitivity.

The trained classifier will be sensitive to the perturbation of targets caused by dropout. We note that our approach is contrary to methods called adversarial example training (Goodfellow et al. (2014b); Miyato et al. (2016)) which train the classifier to be robust to adversarial examples. They utilize input noise which can deceive or change the output of the classifier, and incorporate it to obtain a good classifier. Our ADR method encourages the feature generator to obtain noise-robust target features. However, with regard to the classifier, it is trained to be sensitive to noise. To improve the final accuracy, we learn another classifier $C'$ that is not trained to be sensitive to the noise. $C'$ takes features generated by $G$ and is trained with classification loss on source samples. The loss of $C'$ is not used to update $G$. We compare the accuracy of $C$ and $C'$ in experiments on image classification.

## 4 EXPERIMENTS

### 4.1 EXPERIMENT ON TOY DATA

**Experimental Setting.** In this experiment, we observe the decision boundary obtained by each neuron to demonstrate that ADR encourages the neurons to learn different input characteristics. We use synthetic "two moons" data for this problem. Two dimensional samples from two classes are generated as source samples. Target samples are obtained by rotating the source samples. In our setting, the rotation was set to 30 degrees and data was generated with *scikit-learn* (Pedregosa et al. (2011)). We train a six-layered fully-connected network; the lower 3 layers are used as feature generator, and upper 3 layers are used as classifier. We used Batch Normalization (Ioffe & Szegedy (2015)) and ReLU as activation function. The number of neurons are [2,5,5] for feature generator, [5,5,2] for classifier. We visualize the boundary obtained from each neuron in the last layer by removing the output of all other neurons.

**Results.** We show the learned boundary in Fig. 3. In the baseline model trained only with source samples (top row), two of five neurons do not seem to learn an effective boundary, and three neurons learn a similar boundary. On the other hand, in our method (bottom row), although two neurons do not seem to learn any meaningful boundary, three neurons learn distinctive boundaries, demonstrating greater diversity. Each neuron is trained to be sensitive to the noise caused by target samples. The final decision boundary (rightmost column) classifies most target samples correctly. The accuracy of our proposed method is 96% whereas the accuracy of the non-adapted model was 84%.

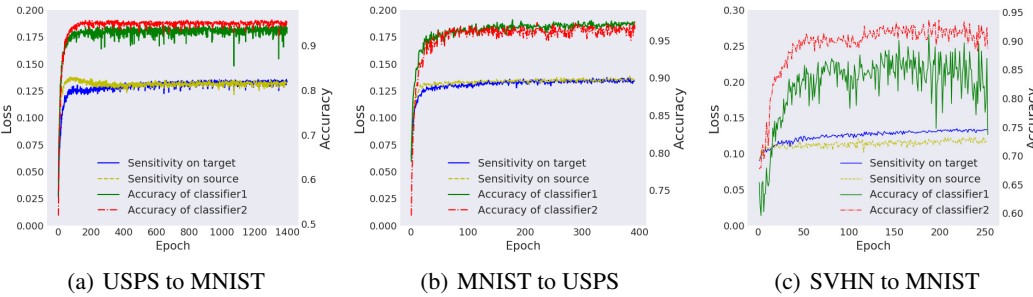

| (a) USPS to MNIST | (b) MNIST to USPS | (c) SVHN to MNIST |

Figure 4: Relationship between sensitivity loss on target (**blue** line), on source (**yellow** line), and accuracy (**red**: accuracy of $C'$, **green**: accuracy of $C$) during training on digits.

## 4.2 Unsupervised Domain Adaptation for Classification

**Experiments on Digits Classification.** We evaluate our model on adaptation between digits datasets. We use MNIST (LeCun et al. (1998)), SVHN (Netzer et al. (2011)) and USPS datasets and follow the protocol of unsupervised domain adaptation used by (Tzeng et al. (2017)). To extensively compare our method with previous methods, in adaptation from MNIST to USPS, we applied our method to a different protocol used in Bousmalis et al. (2017). We assume no labeled target samples and use fixed hyper-parameters for all experiments, unlike other works that use a target validation set (Saito et al. (2017)). The number of iterations for Step 3 was fixed at $n = 4$. We used the same network architecture as in (Tzeng et al. (2017)), but inserted a Batch Normalization layer before the activation layer to stabilize the training. We used Adam (Kingma & Ba (2014)) for optimizer and set the learning rate to $2.0 \times 10^{-4}$, a value commonly reported in the GAN literature. We compare our approach to several existing methods and to the entropy minimization baseline (ENT) obtained by modifying (Springenberg (2015)). As we mentioned in Section 2, this is a model that generates features to minimize the entropy of the output probability for target samples. Due to space limitations, we provide a detailed explanation of this baseline in the appendix.

Results in Table 1 demonstrate that ADR obtains better performance than existing methods. In particular, on the challenging adaptation task from SVHN to MNIST, our method achieves much better accuracy than previously reported. Fig. 4 shows the learning curve of each experiment. As sensitivity loss increases, the target accuracy improves. This means that as critic $C$ learns to detect the non-discriminative samples, feature generator $G$ learns to fool it, resulting in improved accuracy. In addition, we can see that the sensitivity of source samples increases too. As mentioned in Sec 3.3, the critic network should learn to capture features which are not very important for classification, such as texture or slight edges, and it seems to also capture such information in source samples. The accuracy of the classifier $C'$ (denoted by red), which is trained not to be sensitive to the noise, is almost always better than the accuracy of the critic network. In adaptation from SVHN to MNIST (Fig. 5(c)), the accuracy of the critic often suffers as it becomes too sensitive to the noise caused by dropout. On the other hand, the accuracy shown by the red line is stable. Our ENT baseline shows good performance compared to other existing methods. This result indicates the effectiveness of methods based on entropy minimization. In Fig. 5, we compare our proposed method and ENT in terms of entropy of target samples. Our method clearly decreases the entropy, because target samples are moved away from the decision boundary. Yet, its behavior is different from ENT. Interestingly, the entropy is made smaller than ENT in case of adaptation from USPS to MNIST (Fig. 4(a)) though ENT directly minimizes the entropy and our method does not. On the SVHN to MNIST task (Fig. 4(c)), the entropy of ADR is larger than ENT, which indicates that our method places the target samples closer to the decision boundary than ENT does.

**Experiments on Object Classification.** We next evaluate our method on fine-tuning a pretrained CNN. We use a new domain adaptation benchmark called the VisDA Challenge (Peng et al. (2017)) which focuses on the challenging task of adapting from synthetic to real images. The source domain consists of 152,409 synthetic 2D images from 12 object classes rendered from 3D models. The validation and test target domains consists of real images, which belong to the same classes. We used the validation domain (55,400 images) as our target domain in an unsupervised domain adaptation setting.

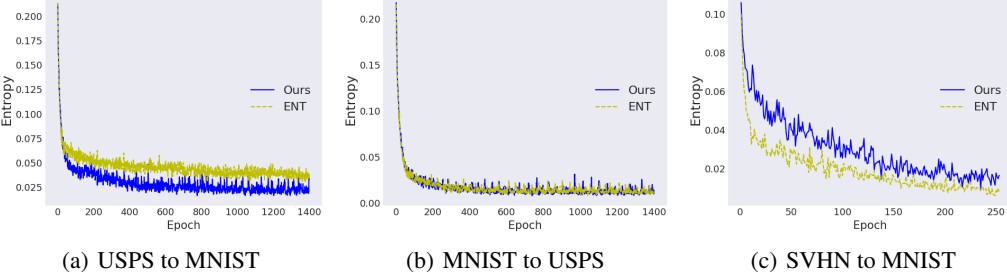

Figure 5: Comparison of entropy of ours (**blue** line) with ENT (**yellow** line). The entropy is calculated on target samples by using the output of the classifier.

| METHOD | SVHN to MNIST | USPS to MNIST | MNIST(P1) to USPS | MNIST(P2) to USPS |
|---|---|---|---|---|
| Source Only | 67.1 | 68.1 | 77.0 | 78.9 |
| LTN (Sener et al. (2016)) | 78.8 | - | - | - |
| ATDA (Saito et al. (2017)) | 86.2† | - | - | - |
| DSN (Bousmalis et al. (2016)) | 82.7† | - | - | 91.3† |
| PixelDA (Bousmalis et al. (2017) | - | - | - | 95.9† |
| DANN (Ganin & Lempitsky (2014)) | 73.9 | 73.0±2.0 | 77.1±1.8 | 85.1† |
| DoC (Tzeng et al. (2014)) | 68.1±0.3 | 66.5±3.3 | 79.1±0.5 | - |
| ADDA (Tzeng et al. (2017)) | 76.0±1.8 | 90.1±0.8 | 89.4±0.2 | - |
| CoGAN (Liu & Tuzel (2016)) | did not converge | 89.1±0.8 | 91.2±0.8 | - |
| DTN (Taigman et al. (2016)) | 84.7 | - | - | - |
| ENT (Our proposed baseline) | 94.9±4.11 | 91.2±1.92 | 93.7±0.54 | 96.7±1.27 |
| Ours | 95.0±1.87 | 93.1±1.27 | 93.2±2.46 | 96.1±0.29 |

Table 1: Results on digits datasets. Please note that † means the result obtained using a few labeled target samples for validation. The reported accuracy of our method is obtained from $C'$. ENT is our proposed entropy minimization baseline, described in the appendix. MNIST(P1) and MNIST(P2) indicate different experimental settings used in Tzeng et al. (2017) and Bousmalis et al. (2017) respectively.

We evaluate our model on fine-tuning networks pretrained on ImageNet (Deng et al. (2009)): ResNet101 (He et al. (2016)) and ResNext (Xie et al. (2016)). For the feature generator, we use the pretrained CNN after removing the top fully connected layer. For the classification network, we use a three-layered fully connected network.

Table 2 shows that our method outperformed other distribution matching methods and our new baseline (ENT) in finetuning both networks by a large margin. ENT did not achieve better performance than existing methods, though improvement over the source only model was observed. Although this method performed well on digits, it does not work as well here, possibly because of the larger shift between very different domains. In the experiment on ResNext, after training $G$ and $C$, we retrained a classifier $C'$ just on the features generated by $G$ due to GPU memory limitations, and observed improvement in both networks.

Fig. 6 visualizes the target features obtained by $G$ with the pretrained model, model fine-tuned on source, and our ADR method. While the embedding of the source only model does not separate classes well due to domain shift, we can see clearly improved separation with ADR.

**Image Segmentation experiments.** Next, we apply our method to adaptation for semantic image segmentation. Image segmentation is different from classification in that we classify each pixel in the image. To evaluate the performance on segmentation, the synthetic GTA5 (Richter et al. (2016)) dataset is used as source, and real CityScape (Cordts et al. (2016)) dataset is used as target. Previous work tackled this problem by matching distributions of each pixel's feature in a middle layer of the network (Hoffman et al. (2016)). In this work, we apply ADR by calculating sensitivity between all pixels. The training procedure is exactly the same as in classification experiments.

We use the ResNet50 pretrained on ImageNet, and utilize an FCN (Long et al. (2015a)) based network architecture. Further, we utilize the more recent Dilated Residual Networks (DRN) 105 layered model (Yu et al. (2017)), which outperforms ResNet50 on a semantic segmentation task.

| Method | aeroplane | bicycle | bus | car | horse | knife | motorcycle | person | plant | skateboard | train | truck | mean |
|---|---|---|---|---|---|---|---|---|---|---|---|---|---|
| **Finetuning on ResNet101** | | | | | | | | | | | | | |
| Source Only | 55.1 | 53.3 | 61.9 | 59.1 | 80.6 | 17.9 | 79.7 | 31.2 | 81.0 | 26.5 | 73.5 | 8.5 | 52.4 |
| MMD | 87.1 | 63.0 | 76.5 | 42.0 | **90.3** | 42.9 | 85.9 | 53.1 | 49.7 | 36.3 | **85.8** | 20.7 | 61.1 |
| DANN | 81.9 | **77.7** | 82.8 | 44.3 | 81.2 | 29.5 | 65.1 | 28.6 | 51.9 | 54.6 | 82.8 | 7.8 | 57.4 |
| ENT | 80.3 | 75.5 | 75.8 | 48.3 | 77.9 | 27.3 | 69.7 | 40.2 | 46.5 | 46.6 | 79.3 | 16.0 | 57.0 |
| Ours | 94.1 | 51.3 | 83.2 | 72.2 | 88.7 | 68.8 | **92.8** | 70.2 | 77.2 | **63.6** | 82.9 | **30.3** | 72.9 |
| Ours (retrained classifier) | **94.2** | 48.5 | **84.0** | 72.9 | 90.1 | **74.2** | 92.6 | 72.5 | 80.8 | 61.8 | 82.2 | 28.8 | **73.5** |
| **Finetuning on ResNeXt** | | | | | | | | | | | | | |
| Source Only | 74.3 | 37.6 | 61.8 | 68.2 | 59.5 | 10.7 | 81.4 | 12.8 | 61.6 | 26.0 | 70.0 | 5.6 | 47.4 |
| MMD | 90.7 | 51.1 | 64.8 | 65.6 | 89.9 | 46.5 | 91.9 | 40.1 | 81.5 | 24.1 | **90.0** | 28.5 | 63.7 |
| DANN | 86.0 | 66.3 | 60.8 | 56.0 | 79.8 | 53.7 | 82.3 | 25.2 | 58.2 | 31.0 | 89.3 | 26.1 | 59.6 |
| ENT | **94.7** | **81.0** | 57.0 | 46.6 | 73.9 | 49.0 | 69.2 | 31.0 | 40.5 | 34.3 | 87.3 | 15.1 | 56.6 |
| Ours | 86.3 | 71.9 | **87.6** | 78.1 | 93.0 | 84.8 | **94.5** | 78.9 | **91.8** | **58.9** | 77.7 | 26.7 | 77.5 |
| Ours (retrained classifier) | 89.2 | 70.9 | 85.7 | **82.0** | **93.7** | **86.7** | 93.3 | 72.3 | 89.5 | 53.0 | 86.7 | 28.3 | **77.6** |

Table 2: Results on Visda2017 classification datasets (Peng et al. (2017)). DANN and MMD are distribution alignment methods proposed by (Ganin & Lempitsky (2014)) and (Long et al. (2015b)) respectively. Ours (retrain classifier) means the classifier retrained for our proposed generator as we mentioned in Sec 3.3. Our proposed method shows much better performance than existing methods.

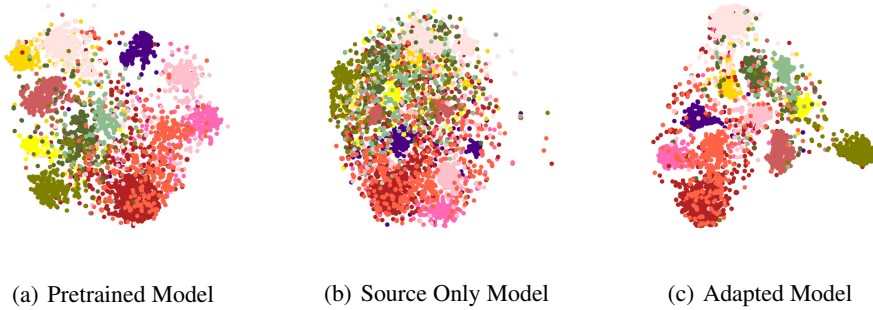

(a) Pretrained Model     (b) Source Only Model     (c) Adapted Model

Figure 6: Visualization of VisDA-classification (12 classes) target features using T-SNE (Maaten & Hinton (2008)): **(a)** features obtained by the Imagenet-pretrained ResNext model not finetuned on VisDA; **(b)** features from the ResNext model fine-tuned only on VisDA source samples without any adaptation; **(c)** features obtained by ResNext adapted by our ADR method.

For the feature generator, we use the pretrained network without fully-connected layers. For the classifier, we use a fully-convolutional network with dropout layers. Due to limited memory, the batch size is set to 1. We include details of the network architecture in appendix. For comparison, we train a domain classifier based model for our network (DANN). We build a domain classifier network for the features of each pixel following (Hoffman et al. (2016)).

In Table 3, we show the qualitative comparison with existing methods. ADR clearly improves mean IoU (Intersection-over-Union) compared to the source-only and competing models, beating state-of-the-art by a large margin. When we apply ADR to DRN, the accuracy improves much more than for ResNet50, and is 12.4 points higher than the model trained only on GTA5 source samples. This is likely because ADR exploits the strong representation of the pretrained DRN network. Although we implemented ENT in this setting, the accuracy was much worse than the Source Only model with a mIoU of 15.0 in training ResNet50. The ENT method does not seem to work well on synthetic-to-real shifts. Finally, we illustrate our method's improvement on example input images, ground truth labels, images segmented by the Source Only model and our method in Fig. 7. While the Source Only model seems to suffer from domain shift, ADR generates a clean segmentation. These experiments demonstrate the effectiveness of ADR on semantic segmentation.

## 5 CONCLUSION

In this paper, we introduced a novel approach for aligning deep representation, Adversarial Dropout Regularization, which learns to generate discriminative features for the target domain. The method

| Network | Method | road | sidewalk | building | wall | fence | pole | t light | t sign | veg | terrain | sky | person | rider | car | truck | bus | train | mbike | bike | mIoU |
|---|---|---|---|---|---|---|---|---|---|---|---|---|---|---|---|---|---|---|---|---|---|
| VGG-16 | FCN Wild | 70.4 | **32.4** | 62.1 | 14.9 | 5.4 | 10.9 | 14.2 | 2.7 | 79.2 | 21.3 | 64.6 | 44.1 | 4.2 | 70.4 | 8.0 | 7.3 | 0.0 | 3.5 | 0.0 | 27.1 |
| ResNet50 | Source Only | 64.5 | 24.9 | 73.7 | 14.8 | 2.5 | 18.0 | 15.9 | 0.0 | 74.9 | 16.4 | 72.0 | 42.3 | 0.0 | 39.5 | 8.6 | 13.4 | 0.0 | 0.0 | 0.0 | 25.3 |
| | DANN | 72.4 | 19.1 | 73.0 | 3.9 | 9.3 | 17.3 | 13.1 | 5.5 | 71.0 | 20.1 | 62.2 | 32.6 | 5.2 | 68.4 | 12.1 | 9.9 | 0.0 | 5.8 | 0.0 | 26.4 |
| | Ours | **87.8** | 15.6 | 77.4 | **20.6** | **9.7** | 19.0 | 19.9 | 7.7 | 82.0 | 31.5 | 74.3 | 43.5 | 9.0 | 77.8 | 17.5 | 27.7 | **1.8** | 9.7 | 0.0 | 33.3 |
| DRN-105 | Source Only | 25.9 | 10.9 | 50.5 | 3.3 | 12.2 | 25.4 | 28.6 | 13.0 | 78.3 | 7.3 | 63.9 | 52.1 | 7.9 | 66.3 | 5.2 | 7.8 | 0.9 | 13.7 | 0.7 | 24.9 |
| | Ours | 86.2 | 10.1 | **78.8** | 20.1 | 7.4 | **21.2** | **26.5** | **15.0** | **84.5** | **38.9** | **81.1** | **54.6** | **13.6** | **80.8** | **29.6** | **32.7** | 0.2 | **19.2** | **8.3** | **37.3** |

Table 3: Results on adaptation from GTA5 → Cityscapes. DANN and FCN Wild denote methods proposed by (Ganin & Lempitsky (2014)) and (Hoffman et al. (2016)) respectively.

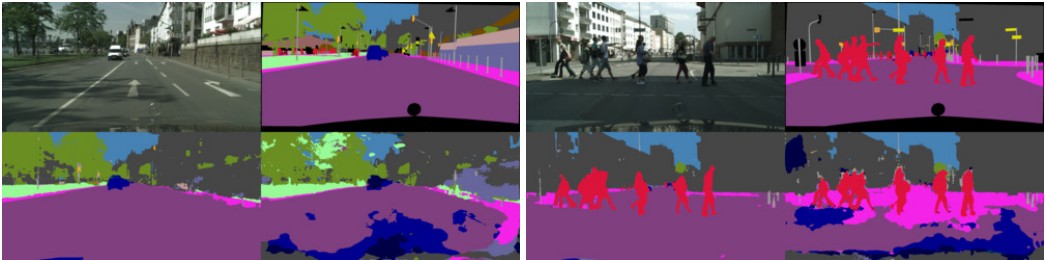

Figure 7: Comparison of results on two real images segmented by ResNet50. Clockwise from upper left: Original image; Ground truth; Segmented image before adaptation; Segmented image after adaptation by our method.

consists of a critic network that can detect samples near the task decision boundary and a feature generator that fools the critic. Our approach is general, applies to a variety of tasks, and does not require target domain labels. In extensive domain adaptation experiments, our method outperformed baseline methods, including entropy minimization, and achieved state-of-the-art results on three datasets.

We also show how to apply our method to train Generative Adversarial Networks for semi-supervised learning in the appendix.

## 6 ACKNOWLEDGEMENTS

We would like to thank Trevor Darrell for his great advice on our paper. The first author's stay at Boston University was partially supported by a scholarship from the University of Tokyo. The work was partially funded by the ImPACT Program of the Council for Science, Technology, and Innovation (Cabinet Office, Government of Japan), and was partially supported by CREST, JST. Saenko was supported by IARPA and NSF grants CCF-1723379 and IIS-1724237.

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

APPENDIX

## A  ENTROPY BASED METHOD FOR DOMAIN ADAPTATION (ENT)

Our method aims to move target samples away from the decision boundary. Some techniques used in training Generative Adversarial Networks can be applied to achieve our goal too. (Springenberg (2015); Salimans et al. (2016)) used small number of labeled samples to train critic. Critic is trained to classify real samples into $K$ classes. They also trained critic to move unlabeled real images away from the boundary by minimizing entropy of the critic's output. Generated fake images are moved near the boundary by maximizing the entropy. On the other hand, generator is trained to generate fake images which should be placed away from the boundary. This kind of method can be easily applied to domain adaptation problem. We would like to describe the method along with our problem setting.

Similar to our method, we have critic networks $C$ and generator $G$. $C$ classifies samples into $K$ class. $C$ is trained to maximize the entropy of target samples, which encourages to move the target samples near the boundary. Then, $G$ is trained to minimize the entropy of them. Thus, $G$ tries to move target samples away from the boundary.

The only difference from our method is that we used entropy term for adversarial training loss. That is, in this method, we replace our sensitivity term $d(p_1, p_2)$ in Eq. 4 with entropy of the classifier output. The adversarial loss for this baseline method is a following one.

$$L_{adv}(X_t) \quad = \quad \mathbb{E}_{\mathbf{x_t} \sim X_t}[H[p(\mathbf{y}|\mathbf{x_t})]] \tag{6}$$

$$H[p(\mathbf{y}|\mathbf{x_t})] \quad = \quad -\sum_{k=1}^{K} p(y = k|\mathbf{x_t}) \log p(y = k|\mathbf{x_t}) \tag{7}$$

$$H[p(y|x)] = -\sum_{k=1}^{K} p(y = k|x) \log p(y = k|x) \tag{8}$$

The hyper-parameter $n$, how many times we update $G$ for adversarial loss in one mini-batch, is set as $n = 4$. Experimentally, it worked well for all settings.

## B  DIGITS CLASSIFICATION TRAINING DETAIL

We follow the protocol used in (Tzeng et al. (2017)). For adaptation from SVHN to MNIST, we used standard training splits of each datasets as training data. For evaluation, we used test splits of MNIST. For the adaptation between MNIST and USPS (P1), we sampled 2000 images from MNIST and 1800 images from USPS. For the adaptation between MNIST and USPS (P2), we used all training images of MNIST and USPS following Bousmalis et al. (2017). In these experiments, we composed the mini-batch half from source and half from target samples. The batch-size was set as 128 for both source and target. We report the score after repeating Step 1∼3 (please see Sec 3.2) 20000 times. For our baseline, ENT, we used the same network architecture and the same hyper-parameters as used in our proposed method.

## C  OBJECT CLASSIFICATION TRAINING DETAIL

In this experiment, SGD with learning rate $1.0 \times 10^{-3}$ is used to optimize the parameters. For the finetuning of ResNet101, we set batch-size as 32. Due to the limit of GPU memory, we set it as 24 in finetuning ResNext model. We report the score after 20 epochs training. In order to train MMD model, we use 5 RBF kernels with the following standard deviation parameters:

$$\sigma = [0.1, 0.05, 0.01, 0.0001, 0.00001] \tag{9}$$

We changed the number of the kernels and their parameters, but we could not observe significant performance difference. We report the performance after 5 epochs. We could not see any improvement after the epoch.

To train a model (Ganin & Lempitsky (2014)), we used two-layered domain classification networks. Experimentally, we did not see any improvement when the network architecture is changed. According to the original method (Ganin & Lempitsky (2014)), learning rate is decreased every iteration. However, in our experiment, we could not see improvement, thus, we fixed learning rate $1.0 \times 10^{-3}$. We report the accuracy after 1 epoch. The accuracy dropped significantly after the first epoch. We assume this is due to the large domain difference between synthetic and real images.

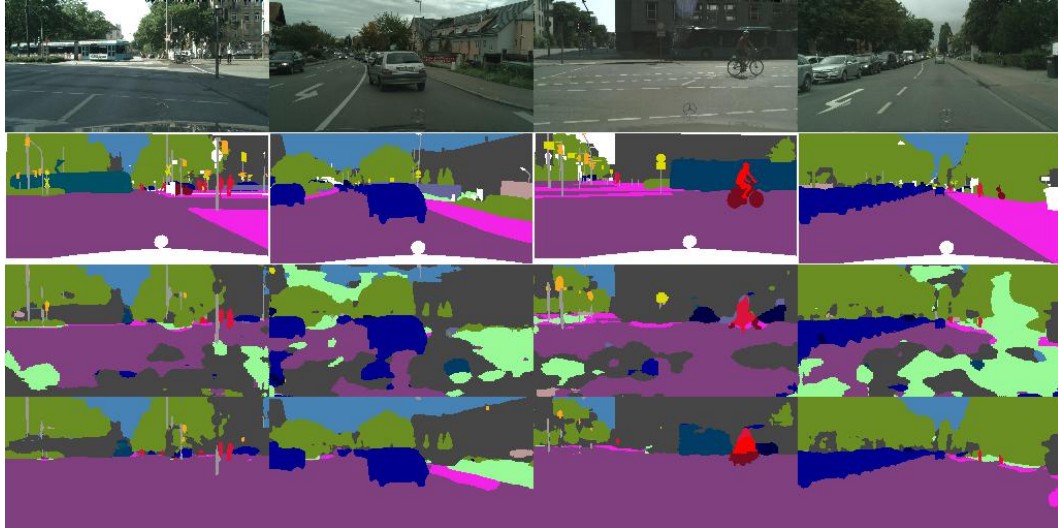

Figure 9: Example of results on segmentation experiments performed by DRN-105. From top to bottom, Original image; Ground truth; Segmented image before adaptation; Segmented image after adaptation by our method.

For our new baseline, ENT, we used the same hyper-parameter as we used for our proposed method. Since the accuracy of ENT drops significantly after around 5 epochs, we report the accuracy after 5 epoch updates.

## D  SEGMENTATION EXPERIMENTS

We modified FCN Long et al. (2015a) architecture suitable for ResNet structure. The features from ResBlock 2∼4 and the first convolution layer and maxpooling layer are used in our implementation. In Fig. 8, we show how we integrated the features of each layers. We regard the layers of ResNet50 as generator and rest of the networks, namely convolution and upsampling layers as a critic network. The input images were resized to 512x1024 due to the limit of GPU memory. For the same reason, the batch-size was set to one. In Fig. 9, we show the example of segmented images by DRN-105. The images are cleanly segmented by our proposed method.

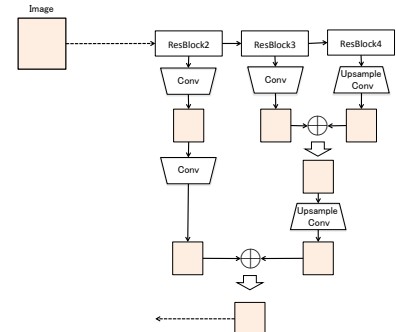

Figure 8: Overview of architecture for semantic segmentation

## E  SEMI-SUPERVISED LEARNING USING GANS

In this section, we demonstrate how to apply our method in training a Generative Adversarial Network (GAN) applied to semi-supervised learning. We follow the method proposed by (Springenberg (2015); Salimans et al. (2016)), who use a $K$-class classification network as a critic to train a GAN in the semi-supervised setting.

**Approach.** In contrast to the domain adaptation setting, here $G$ tries to generate images which fool the critic $C$. Also, in this setting, we are given labeled and unlabeled real images from the same domain. Then, we train the critic to classify labeled images correctly and to move unlabeled images far from the decision boundary. To achieve this, we propose to train the critic with the following objective:

$$\min_C L_C = L(X_L, Y_L) + L_{adv}(X_u) - L_{adv}(X_g) - H[\frac{1}{M}\sum_{i=1}^{M} p(y|x_u{}^i, C)] \tag{10}$$

$$L_{adv}(X_u) = \mathbb{E}_{\mathbf{x_u} \sim X_u}[d(C_1(G(\mathbf{x_u})), C_2(G(\mathbf{x_u})))] \tag{11}$$

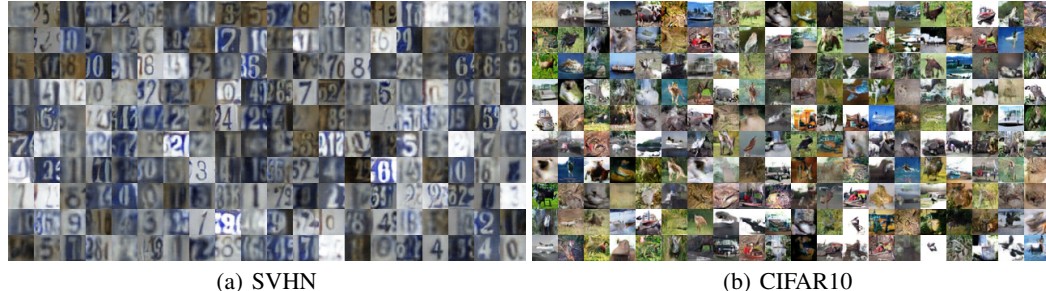

(a) SVHN                                    (b) CIFAR10

Figure 10: Examples of generated images.

|  | SVHN (% errors) | CIFAR (% errors) |
|---|---|---|
| Labeled Only |  |  |
| SDGM (Maaløe et al. (2016) | $16.61 \pm 0.24$ | - |
| CatGAN (Springenberg (2015)) | - | $19.58\pm0.46$ |
| ALI (Dumoulin et al. (2016)) | $7.42\pm0.65$ | **17.99**$\pm1.62$ |
| ImpGAN (Salimans et al. (2016)) | $8.11\pm1.3$ | $18.63\pm2.32$ |
| Ours | **6.26**$\pm1.05$ | $19.63\pm0.37$ |

Table 4: Comparison with state-of-the-art methods on two benchmark datasets. Only methods without data augmentation are included. We used the same critic architecture as used in ImpGAN.

$$L_{adv}(X_g) = \mathbb{E}_{\mathbf{x_g}\sim X_G}[d(C_1(G(\mathbf{x_g})), C_2(G(\mathbf{x_g})))] \qquad (12)$$

where $X_L$ denotes the subset of labeled samples, $X_u$ denotes unlabeled ones and $X_g$ denotes images generated by $G$ and $H$ denotes entropy as Eq.6 shows. The critic is trained to minimize the loss on labeled samples in the first term. Since unlabeled images should be far away from the decision boundary and should be distributed uniformly among the classes, we add the second and fourth term. The third term encourages the critic to detect fake images generated near the boundary.

The objective of $G$ is as follows,

$$\min_G L_{adv}(X_g) + ||\mathbb{E}_{x_g\sim X_g}f(\mathbf{x_g}) - \mathbb{E}_{\mathbf{x_u}\sim X_u}f(\mathbf{x_u})||^2 \qquad (13)$$

where the second term encourages generated images to be similar to real images, which is known to be effective to stabilize the training. The first term encourages the generator to create fake images which should be placed far away from the boundary. Such images should be similar to real images because they are likely to be assigned to some class with high probability. Here, we update $C$ and $G$ same number of times.

**Experiment.** We evaluate our proposed GAN training method by using SVHN and CIFAR10 datasets, using the critic network architecture from (Salimans et al. (2016)). We set the batch size as 100 and used Adam with learning rate $2.0 \times 1.0^{-4}$ for optimizer. After the conv6 layer of the critic, we constructed a classifier which was not concerned with adversarial learning process.

In the experiment on SVHN, we replaced Weight Normalization with Batch Normalization for $C$. Also, in the experiment on CIFAR10, we construct a classifier from a middle layer of the critic, which is not incorporated into the adversarial training step. This is motivated by the insight that the critic in our method is trained to be too sensitive to the dropout noise as we explained in Sec 3.3.

**Results.** From Fig. 10(a), we can see that ADR seems to generate realistic SVHN images. Some images are significantly blurred, but most of the images are clear and diverse. As for generated CIFAR10 images, they do not seem as realistic, but some objects appear in most images. In Table 4, we can see that the accuracy of the critic trained by our method has better performance than other models for SVHN. For CIFAR10, the accuracy was slightly worse than other state-of-the-art methods. We conclude that, despite its clear advantage on the domain adaptation tasks, our method produces mixed results on the SSL tasks. It could still be useful for SSL, however, it needs further exploration to improve the accuracy. For example, in Eq. 6, we propose to maximize the entropy of

the marginal class distribution of the unlabeled real images, as well as forcing them to be far from the boundary. However, these objectives may contradict each other, which may in turn degrade the performance. In late-breaking results, Dai et al. (2017) theoretically showed that just generating fake images that are far from decision boundaries does not help to improve accuracy in training GANs in the setting of SSL. Further improvement of our SSL approach based on these results is an interesting direction for future work.

