# OpenReview forum: "Adversarial Dropout Regularization"
_ICLR.cc/2018/Conference — Accept (Poster)_

### Official Review · AnonReviewer1 · 2017-11-27

**Rating:** 5
**Confidence:** 4

**Review:**

(Summary)
This paper is about learning discriminative features for the target domain in unsupervised DA problem. The key idea is to use a critic which randomly drops the activations in the logit and maximizes the sensitivity between two versions of discriminators.

(Pros)
The approach proposed in section 3.2 uses dropout logits and the sensitivity criterion between two softmax probability distributions which seems novel.

(Cons)
1. By biggest concern is that the authors avoid comparing the method to the most recent state of the art approaches in unsupervised domain adaptation and yet claims "achieved state of the art results on three datasets." in sec5. 1) Unsupervised Pixel-Level Domain Adaptation with Generative Adversarial Networks, Bousmalis et al. CVPR17, and 2) Learning Transferrable Representations for Unsupervised Domain Adaptation, Sener et al. NIPS16. Does the proposed method outperform these state of the art methods using the same network architectures?
2. I suggest the authors to rewrite the method section 3.2 so that the loss function depends on the optimization variables G,C. In the current draft, it's not immediately clear how the loss functions depend on the optimization variables. For example, in eqns 2,3,5, the minimization is over G,C but G,C do not appear anywhere in the equation.
3. For the digits experiments, appendix B states "we used exactly the same network architecture". Well, which architecture was it?
4. It's not clear what exactly the "ENT" baseline is. The text says "(ENT) obtained by modifying (Springenberg 2015)". I'd encourage the authors to make this part more explicit and self-explanatory.

(Assessment)
Borderline. The method section is not very well written and the authors avoid comparing the method against the state of the art methods in unsupervised DA.

---

> ### Author Response · Authors · 2017-12-18
> **Response to Reviewer1**
>
> We uploaded an updated version of the paper with changes highlighted in blue.
>
> To Reviewer 1
> 1. By biggest concern is that the authors avoid comparing the method to the most recent state of the art approaches in unsupervised domain adaptation and yet claims "achieved state of the art results on three datasets." in sec5. 1) Unsupervised Pixel-Level Domain Adaptation with Generative Adversarial Networks, Bousmalis et al. CVPR17, and 2) Learning Transferrable Representations for Unsupervised Domain Adaptation, Sener et al. NIPS16. Does the proposed method outperform these state of the art methods using the same network architectures?
>
> In the updated version of our paper, we added new experimental results following  the same setting as Bousmalis did (Table 1). Ours is slightly better on MNIST->USPS, but Bousmalis et al. don’t report on more difficult shifts where we achieve state of the art, as such SVHN->MNIST. In addition, we compared our method with Sener et al. NIPS16 in Table 1.
>
> Changes in the Paper
> In Table 1, We  added Sener NIPS16, for SVHN to MNIST. We also added results on MNIST to USPS to compare with Bousmalis CVPR 2016. Results of our method changed in the adaptation using USPS because we found a bug in preprocessing of USPS. According to the change, we replaced the graph of Fig4 (a)(b) and we changed the relevant sentences.
>
> 2. I suggest the authors to rewrite the method section 3.2 so that the loss function depends on the optimization variables G,C. In the current draft, it's not immediately clear how the loss functions depend on the optimization variables. For example, in eqns 2,3,5, the minimization is over G,C but G,C do not appear anywhere in the equation.
>
> We clarified notation of Eqns 2,3,5.
>
> Change of paper
> Change notation of Eqns 2,3,5.
>
> 3. For the digits experiments, appendix B states "we used exactly the same network architecture". Well, which architecture was it?
>
> We wanted to say that, for our baseline method, we used the same network architecture as our proposed method. We added this explanation.
>
> Change of paper.
> Add sentence in the last of our appendix section (Digits Classification Training Detail).
>
> 4. It's not clear what exactly the "ENT" baseline is. The text says "(ENT) obtained by modifying (Springenberg 2015)". I'd encourage the authors to make this part more explicit and self-explanatory.
>
> We did explain it in the appendix, but we added sentences to make the method clearer.
>
> Change of paper
> Add sentence in Section 2, Section 4.2.

---

### Official Review · AnonReviewer2 · 2017-11-28
**An interesting method for domain adaptation**

**Rating:** 7
**Confidence:** 3

**Review:**


Unsupervised Domain adaptation is the problem of training a classifier without labels in some target domain if we have labeled data from a (hopefully) similar dataset with labels. For example, training a classifier using simulated rendered images with labels, to work on real images.
Learning discriminative features for the target domain is a fundamental problem for unsupervised domain adaptation. The problem is challenging (and potentially ill-posed) when no labeled examples are given in the target domain. This paper proposes a new training technique called ADR, which tries to learn discriminative features for the target domain. The key idea of this technique is to move the target-domain features away from the source-domain decision boundary. ADR achieves this goal by encouraging the learned features to be robust to the dropout noise applied to the classifier.

My main concern about this paper is that the idea of "placing the target-domain features far away from the source-domain decision boundary" does not necessarily lead to *discriminative features* for the target domain. In fact, it is easy to come up with a counter-example: the target-domain features are far from the *source-domain* decision boundary, but they are all (both the positive and negative examples) on the same side of the boundary, which leads to poor target classification accuracy. The loss function (Equations 2-5) proposed in the paper does not prevent the occurrence of this counter-example.

Another concern comes from using the proposed idea in training a GAN (Section 4.3). Generating fake images that are far away from the boundary (as forced by the first term of Equation 9) is somewhat opposite to the objective of GAN training, which aims at aligning distributions of real and fake images. Although the second term of Equation 9 tries to make the generated and the real images similar, the paper does not explain how to properly balance the two terms of Equation 9. As a result, I am worried that the proposed method may lead to more mode-collapsing for GAN.

The experimental evaluation seems solid for domain adaptation. The semi-supervised GANs part seemed significantly less developed and might be weakening rather than strengthening the paper.

Overall the performance of the proposed method is quite well done and the results are encouraging, despite the lack of theoretical foundations for this method.

---

> ### Author Response · Authors · 2017-12-18
> **Response to Reviewer 2**
>
> We uploaded an updated version of the paper with changes highlighted in blue.
>
> To Reviewer 2
> 1., My main concern about this paper is that the idea of "placing the target-domain features far away from the source-domain decision boundary" does not necessarily lead to *discriminative features* for the target domain. In fact, it is easy to come up with a counter-example: the target-domain features are far from the *source-domain* decision boundary, but they are all (both the positive and negative examples) on the same side of the boundary, which leads to poor target classification accuracy. The loss function (Equations 2-5) proposed in the paper does not prevent the occurrence of this counter-example.
>
> Yes, we understand that there can be such a counter-example with our method. Note that we add a term that discourages target examples from being placed on one side of the boundary. However it is possible in theory that positive and negative examples switch labels, but we find that this does not occur in practice, and our method works well based on our experimental results.
>
> 2., Another concern comes from using the proposed idea in training a GAN (Section 4.3). Generating fake images that are far away from the boundary (as forced by the first term of Equation 9) is somewhat opposite to the objective of GAN training, which aims at aligning distributions of real and fake images. Although the second term of Equation 9 tries to make the generated and the real images similar, the paper does not explain how to properly balance the two terms of Equation 9. As a result, I am worried that the proposed method may lead to more mode-collapsing for GAN.
> The experimental evaluation seems solid for domain adaptation. The semi-supervised GANs part seemed significantly less developed and might be weakening rather than strengthening the paper.
>
> If the goal is to train a GAN to mimic a distribution only, then our additional objective may not help, but if the goal is to learn features for semi-supervised learning, then our objective helps by forcing the GAN to not generate fake images near the boundary (ambiguous features).

---

> > ### Public Comment · (anonymous) · 2017-12-27
> > **"Forcing the GAN to not generate fake images near the boundary (ambiguous features)" will not help semi-supervised tasks**
> >
> > "If the goal is to train a GAN to mimic a distribution only, then our additional objective may not help, but if the goal is to learn features for semi-supervised learning, then our objective helps by forcing the GAN to not generate fake images near the boundary (ambiguous features)."
> >
> > The authors proposed to add their regularization term to (K+1)-class discriminator formulation of GAN-based SSL such as CatGAN (Springenberg 2015) and Improved GAN (Salimans et al. 2016). They argued that generated fake images away from the boundary can help SSL. However, recently there is some theoretical analysis in BadGAN(Dai et al. 2017) proving that improving the generalization over SSL need a "bad" generator to generate fake images near the boundary. It seems the conclusions of BadGAN and this paper are totally contradictory. Since BadGAN has some theoretical proofs to support their claim, could the authors give some proofs or analysis of why ADR  may help SSL-GAN using fake images not near the boundary? Thanks.
> >
> > Ref
> > Good Semi-supervised Learning That Requires a Bad GAN (NIPS 2017)

---

> > > ### Author Response · Authors · 2018-01-05
> > > **Response to "The wrong objective and ...." and "Forcing the GAN to not ..."**
> > >
> > > As comment "The wrong objective and ...." tells, the objective term should maximize the entropy, not minimize it. The notation was wrong, but our implementation of the experiment was not incorrect. We added more discussion on whether our method works well for SSL tasks. As comment 2 tells, the objective we used in SSL can contradict with the other objective, which may degrade the performance of our method. We considered this point and changed some parts of our paper in SSL.
> > >
> > > With regard to comment "Forcing the GAN to not ...", we do not have a theoretical analysis of why ADR may help SSL-GAN. From the results of the SSL experiments, we cannot conclude that our method is better than other state-of-the-art methods for SSL. We also need further theoretical analysis and improvement to construct a method that works well on SSL, but we have not yet. We changed our paper to emphasize this point.
> > >
> > > Revised parts are indicated by red characters.

---

### Official Review · AnonReviewer3 · 2017-12-01
**Fresh idea on adversarial training for domain adaptation**

**Rating:** 8
**Confidence:** 5

**Review:**

I think the paper was mostly well-written, the idea was simple and great. I'm still wrapping my head around it and it took me a while to feel convinced that this idea helps with domain adaptation. A better explanation of the intuition would help other readers. The experiments were extensive and show that this is a solid new method for trying out for any adaptation problem. This also shows how to better utilize task models associated with GANs and domain adversarial training, as used eg. by Bousmalis et al., CVPR 2017, or Ganin et al, ICML 2015, Ghifary et al, ECCV 2016, etc.

I think important work was missing in related work for domain adaptation. I think it's particularly important to talk about pixel/image-level adaptations eg CycleGAN/DiscoGAN etc and specifically as those were used for domain adaptation such as Domain Transfer Networks, PixelDA, etc. Other works like Ghifary et al, 2016, Bousmalis et al. 2016 could also be cited in the list of matching distributions in hidden layers of a CNN.

Some specific comments:

Sect. 3 paragraph 2 should be much clearer, it was hard to understand.

In Sect. 3.1 you mention that each node of the network is removed with some probability; this is not true. it's each node within a layer associated with dropout (unless you have dropout on every layer in the network).  It also wasn't clear to me whether C_1 and C_2 are always different. If so, is the symmetric KL divergence still valid if it's minimizing the divergence of distributions that are different in every iteration? (Nit: capitalize Kullback Leibler)

Eq.3 I think the minus should be a plus?

Fig.3 should be improved, it wasn't well presented and a few labels as to what everything is could help the reader significantly. It also seems that neuron 3 does all the work here, which was a bit confusing to me. Could you explain that?

On p.6 you discuss that you don't use a target validation set as in Saito et al. Is one really better than the other and why? In other words, how do you obtain these fixed hyperparameters that you use?

On p. 9 you claim that the unlabeled images should be distributed uniformly among the classes. Why is that?

---

> ### Author Response · Authors · 2017-12-18
> **Response to Reviewer 3**
>
> We uploaded an updated version of the paper with changes highlighted in blue.
>
> To Reviewer 3
> 1, I think important work was missing in related work for domain adaptation. I think it's particularly important to talk about pixel/image-level adaptations eg CycleGAN/DiscoGAN etc and specifically as those were used for domain adaptation such as Domain Transfer Networks, PixelDA, etc. Other works like Ghifary et al, 2016, Bousmalis et al. 2016 could also be cited in the list of matching distributions in hidden layers of a CNN.
>
> We will refer to such methods and compare with PixelDA as possible as we can. (Same question as Reviewer1, 1)
>
> 2. Sect. 3 paragraph 2 should be much clearer, it was hard to understand.
>
> We changed paragraph 2 of section 3.
>
> 3. In Sect. 3.1 you mention that each node of the network is removed with some probability; this is not true. it's each node within a layer associated with dropout (unless you have dropout on every layer in the network).  It also wasn't clear to me whether C_1 and C_2 are always different. If so, is the symmetric KL divergence still valid if it's minimizing the divergence of distributions that are different in every iteration? (Nit: capitalize Kullback Leibler)
> ”It also wasn't clear to me whether C_1 and C_2 are always different”
>
> →C_1 and C_2 are not necessarily always different. C_1 and C_2 can be the same classifier. However, it rarely happens.
>  “If so, is the symmetric KL divergence still valid if it's minimizing the divergence of distributions that are different in every iteration?”
> →Yes, we think it is valid. The generator tries to minimize the divergence. The divergence means the sensitivity to noise caused by dropout. The goal of minimizing it is to generate features that are insensitive to the dropout noise. We minimize the divergence of distributions that are different in almost every iteration.
>
> 4. Eq.3 I think the minus should be a plus?
>
> No. In Eq.3, we aim to maximize the sensitivity for classifiers. In this phase, the classifiers should be trained to be sensitive to the noise caused by dropout. Thus, the minus should be a minus.
>
> 5. Fig.3 should be improved, it wasn't well presented and a few labels as to what everything is could help the reader significantly. It also seems that neuron 3 does all the work here, which was a bit confusing to me. Could you explain that?
>
> We improved the presentation. Neuron 3 seems to be dominant in bottom row (our method. However, when comparing Neuron 3 and Column 6, the shape of boundary looks a little different because of the effect of other neurons. What we wanted to show here is that each neurons will learn different features by our method. We will improve our presentation.
>
> Change of paper
> Add notation in Figure 3, add caption.
>
> 6., On p.6 you discuss that you don't use a target validation set as in Saito et al. Is one really better than the other and why? In other words, how do you obtain these fixed hyperparameters that you use?
>
>
> The main hyperparameter in our method is n, which indicates how many times to repeat Step 3 in our method. We set 4 in our experiments. Although we did not show in our experimental results, we tried other number such as 1,2,3. Through the experiment, we found that 4 works well in most settings. With regard to other hyperparameters, such as batch-size, learning rate, we used the ones that are common in other papers on domain adaptation.
> If one uses a target val set (as in Saito et al.), then one assumes access to training labels on target, which we don’t want to assume in our setting.
>
> 7. On p. 9 you claim that the unlabeled images should be distributed uniformly among the classes. Why is that?
>
> We assumed that it is not desirable if unlabeled images are aligned with one class. We add this term following “Unsupervised and semi-supervised learning with categorical generative adversarial networks”.

---

> > ### Public Comment · (anonymous) · 2017-12-27
> > **The wrong objective and wrong claim of " distributed uniformly among the classes" in the semi-supervised learning part**
> >
> > * On p. 9 you claim that the unlabeled images should be distributed uniformly among the classes.
> > First, the last term E_{x_u}\sum_{k=1}^{K}p(y=k|x_u)\log p(y|x_u) in  your objective in Eq.(6)
> > min_C L(X_L,Y_L) + L_{adv}(X_u)- L_{adv}(X_g) + E_{x_u}\sum_{k=1}^{K}p(y=k|x_u)\log p(y|x_u)
> > is the negative entropy term of each unlabeled data. Minimizing it (equivalently, maximize the entropy) can NOT achieve "distributed uniformly among the classes" as you claimed. The correct way is to maximize the entropy of the marginal class distribution H(E_{x_u}p(y|x_u)) = H(\frac{1}{N}\sum_{i=1}^{N}p(y|x^{i}_u)) as shown in Eq.(6) in the paper CatGAN( "Unsupervised and semi-supervised learning with categorical generative adversarial networks").
> > In fact, this term enforces the unlabeled data to locate near the decision boundary, which contradicts with the second term L_{adv}(X_u). Then the objective is self-contradictory, i.e., the second term is to make unlabeled data far away from the boundary and the last term is to make them near the boundary. Thus the performance is worse than your baseline (ImprovedGAN) on CIFAR-10, and far from the state-of-the-art results, e.g.  NIPS 2017 (Good semi-supervised learning that requires a bad gan).

---

> > > ### Author Response · Authors · 2018-01-05
> > > **Response**
> > >
> > > Thank you for finding and pointing out this error in our notation! Please see our detailed response below in the comment titled: Response to "The wrong objective and ...." and "Forcing the GAN to not ..."

---

> > > > ### Public Comment · (anonymous) · 2018-01-06
> > > > **The response is still wrong and it seems you misunderstand what the error is**
> > > >
> > > > Your response below is ***As comment "The wrong objective and ...." tells, the objective term should maximize the entropy, not minimize it. The notation was wrong, but our implementation of the experiment was not incorrect. ***
> > > >
> > > > The error is NOT "minimize the entropy", actually you indeed maximized the entropy of p(y|x_u) in the previous version because you minimize the negative of the entropy (min_C E_{x_u}\sum_{k=1}^{K}p(y=k|x_u)\log p(y|x_u) equals to max_C -E_{x_u}\sum_{k=1}^{K}p(y=k|x_u)\log p(y|x_u)).
> > > > Your error that I pointed out is you used the wrong objective, which did not correspond to your claim "distributed uniformly among the classes". Your objective maximizes E_{x_u} H(p(y|x_u)) but the correct way to achieve "distributed uniformly among the classes" is to maximize the MARGINAL ENTROPY H(E_{x_u} p(y|x_u)) (see Eq.(6) in the paper CatGAN).
> > > >
> > > > By the way, the revised objective in the paper is still not correct now (the sum over k is wrong). I do not believe your response "our implementation of the experiment was not incorrect" because your explanation of the error is a misunderstanding.

---

> > > > > ### Author Response · Authors · 2018-01-06
> > > > > **Fixed wrong parts of our paper**
> > > > >
> > > > > We have double checked our implementation and it was icorrect, so the error was only in the equation written in the original paper draft.  Thus the notation error did not affect our experiments.
> > > > >
> > > > > The codes are here. We are using Pytorch.
> > > > > This is the minimized objective.
> > > > > def entropy(self,output):
> > > > >       prob = F.softmax(output)
> > > > >       prob_m = torch.mean(prob,0)
> > > > >       return torch.sum(prob_m*torch.log(prob_m+1e-6))
> > > > > prob is the output of the classifier, which is a matrix of MxC dimension.
> > > > > M indicates the number of samples in mini-batch and C is the number of classes.
> > > > > Thus, we first calculate the marginal class probability. Then, we calculate the entropy

---

### Public Comment · (anonymous) · 2017-12-27
**Comparison to recent state-of-the-art approaches in unsupervised domain adaptation**

There are a lot of works on domain adaptation this year. For example, self-ensembling for domain adaptation (https://arxiv.org/pdf/1706.05208.pdf). And their results seem much better. It would be better to include these methods for comparison.

---

> ### Author Response · Authors · 2018-01-05
> **Response to comment on comparison**
>
> The paper you are referring to has not been accepted by any peer-reviewed conference or journal, and has only been posted very recently on arxiv. Therefore, we should not be obligated to compare to their reported results. We do include thorough comparisons with many recent methods in our paper.  Also, their method utilizes various data augmentation, which we did not do in most settings (in adaptation for VisDA, we conducted random crops and flipping).

---

### Decision · Program_Chairs · 2018-01-29
**ICLR 2018 Conference Acceptance Decision**

**Decision:**

Accept (Poster)

**Comment:**

The general consensus is that this method provides a practical and interesting approach to unsupervised domain adaptation. One reviewer had concerns with comparing to state of the art baselines, but those have been addressed in the revision.

There were also issues concerning correctness due to a typo. Based on the responses, and on the pseudocode, it seems like there wasn't an issue with the results, just in the way the entropy objective was reported.

You may want to consider reporting the example given by reviewer 2 as a negative example where you expect the method to fail. This will be helpful for researchers using and building on your paper.